# Researching New Drug Combinations with Senolytic Activity Using Senescent Human Lung Fibroblasts MRC-5 Cell Line

**DOI:** 10.3390/ph17010070

**Published:** 2024-01-04

**Authors:** Maria Carolina Ximenes de Godoy, Juliana Alves Macedo, Alessandra Gambero

**Affiliations:** 1School for Life Sciences, Pontifical Catholic University of Campinas (PUC-Campinas), Av. John Boyd Dunlop, s/n, Campinas 13034-685, SP, Brazil; maria.cxg@puccampinas.edu.br; 2Department of Food and Nutrition, School of Food Engineering, State University of Campinas, Campinas 13083-862, SP, Brazil; jumacedo@unicamp.br

**Keywords:** dasatinib, quercetin, ellagic acid, resveratrol, senotherapeutic, cell senescence

## Abstract

Therapeutically targeting senescent cells seems to be an interesting perspective in treating chronic lung diseases, which are often associated with human aging. The combination of the drug dasatinib and the polyphenol quercetin is used in clinical trials as a senolytic, and the first results point to the relief of physical dysfunction in patients with idiopathic pulmonary fibrosis. In this work, we tested new combinations of drugs and polyphenols, looking for senolytic activity using human lung fibroblasts (MRC-5 cell line) with induced senescence. We researched drugs, such as azithromycin, rapamycin, metformin, FK-506, aspirin, and dasatinib combined with nine natural polyphenols, namely caffeic acid, chlorogenic acid, ellagic acid, ferulic acid, gallic acid, epicatechin, hesperidin, quercetin, and resveratrol. We found new effective senolytic combinations with dasatinib and ellagic acid and dasatinib and resveratrol. Both drug combinations increased apoptosis, reduced BCL-2 expression, and increased caspase activity in senescent MRC-5 cells. Ellagic acid senolytic activity was more potent than quercetin, and resveratrol counteracted inflammatory cytokine release during senolysis in vitro. In conclusion, dasatinib and ellagic acid and dasatinib and resveratrol present in vitro senolytic potential like that observed for the combination in clinical trials of dasatinib and quercetin, and maybe they could be future alternatives in the senotherapeutic field.

## 1. Introduction

Aging is the leading risk factor for establishing chronic lung diseases, such as lung cancer, chronic obstructive pulmonary disease (COPD), and idiopathic pulmonary fibrosis (IPF). In addition to chronological aging, other factors, such as exposure to environmental pollutants, smoking, drugs, stress, and infections accelerate the aging process of lung cells, a phenomenon known as cellular senescence, and the risk of developing chronic lung diseases [1].

Usually, early senescent cells are eliminated from the tissue environment by the action of the immune system. However, when senescence is established in the immune system cells, the removal of senescent cells is compromised, which generates their accumulation in tissues and the induction of senescence in neighboring cells [2]. Morphological and structural changes, such as multinucleated cells with enlarged vacuoles and increased and flattened morphology, are commonly observed in senescent cells [3]. Senescent cells are characterized by activation of p16^ink4a^/pRB and p53/p21WAF1/CIP1 tumor suppressor pathways which is associated with cell cycle arrest [4], by resistance to apoptosis through upregulation of anti-apoptotic pathways, such as BCL-2, BCL-XL, HSP-90 proteins. or caspase inactivity [5], by positive staining of senescence-associated β-galactosidase (SA-βgal) [6], and by the senescence-associated secretor phenotype (SASP). Although cellular senescence determines a cell cycle arrest that prevents the spread of damage to the next cellular generation and prevents potential malignant transformation, our current understanding is that factors released by SASP may have deleterious functions by exerting autocrine and paracrine-causing effects, such as inflammation, fibrosis, stem cell dysfunction and senescence of neighboring cells [7]. Studies have described the increased presence of fibroblasts and senescent epithelial cells in the lungs of patients with IPF compared to age-matched controls [2,8,9,10]. The repertoire of SASP-associated molecules produced by fibroblasts from patients with IPF includes pro-inflammatory cytokines (such as TNF-α, TGF-β, IL-1β, IL-6, IL-8, IL-10, and IL-18), chemokines (such as CXCL1 and MCP-1), growth regulators (such as FGF, CTGF, GM-CSF, M-CSF, and PDGF), matrix metalloproteinases (MMP-2, MMP-3, MMP-9, MMP- 10, and MMP-12), and leukotrienes (LTA4, LTB4, LTC4, and LTD4) [11,12,13,14].

In COPD, lung fibroblasts display increased markers of cellular senescence compared to age-matched controls, which correlate with loss of lung function. Proteomic analysis of fibroblasts from COPD patients revealed an increased expression of 42 proteins associated with SASP when compared to fibroblasts from patients of the same age, and these were associated with functions related to chronic inflammation, such as cytokines (IL12B, TNFSF14, and RANKL) and chemokines (CCL15, CCL23, and CXCL9) [15]. Fibroblasts from COPD patients also described the secretion of elevated levels of IL-6 and IL-8 and a pro-fibrotic phenotype [10,16]. These data suggest that although COPD and IPF have different clinical behaviors and physiopathology, cellular senescence may be a common therapeutic target.

Therapies that target cellular senescence, including senolytic and senomorphic drugs, stem cell therapies, redox control, and other interventions, have been shown to reduce tissue injury in animal models of lung disease [9,17,18]. Senomorphic drugs suppress the expression of senescence markers and alter the release of cytokines, chemokines, and growth factors associated with SASP without inducing the apoptosis of senescent cells. The senomorphics described so far include inhibitors of IkB kinase (IKK) and nuclear factor (NF)-kB [19], inhibitors of the Janus kinase (JAK) pathway [20], and antioxidant substances, such as polyphenols from natural sources [21]. Some compounds, such as fisetin, a natural polyphenol, have shown senomorphic effects in vitro on some cells while showing senolytic activity on other cell types [22].

Early clinical trials confirm that senotherapeutic approaches using dasatinib plus quercetin could also benefit chronic lung human diseases [23]. The senolytic drug cocktail of small molecules formed by dasatinib, a tyrosine kinase inhibitor, and quercetin, a natural polyphenol, has been shown to induce apoptosis in senescent cells efficiently without any effect on quiescent, proliferating, or differentiated fibroblasts [24]. As mentioned before, another natural polyphenol, fisetin, also has senolytic activity described. We carried out this work looking for new drugs and polyphenol cocktails that could have senolytic activity in human lung fibroblasts with induced senescence. We researched drugs with senotherapeutic activity for senolytic activity screening in combination with different polyphenols. Azithromycin, a macrolide antibiotic, effectively eliminated 44% of senescent myofibroblasts at non-cytotoxic doses for control cells, acting as a senolytic drug by a mechanism related to autophagic and metabolic changes [25]. The mTOR inhibitor rapamycin has senolytic activity by itself, and it is also able to sensitize senolytic cell death induced by navitoclax, suggesting that simultaneous inhibition of mTORC and Bcl-2 could be a new idea for senolytic drug combinations [26,27]. Aspirin, a cyclooxygenase inhibitor mainly used as an antiplatelet drug, reduces cell viability and decreases the levels of Bcl-xL protein in senescent fibroblast [28]. Metformin, a cost-effective and safe biguanide derivative used for the treatment of type 2 diabetes, and FK-506 or tacrolimus, a calcineurin inhibitor, were described as senomorphic drugs due to their ability to control SASP or improve the lifespan of senescent cells, respectively [29,30]. Therefore, azithromycin, rapamycin, aspirin, metformin, and FK-506, in addition to dasatinib, were combined with nine natural polyphenols. The selected polyphenols are chemically diverse, including phenolic acids, such as caffeic acid, chlorogenic acid, ellagic acid, ferulic acid, and gallic acid, flavonoids, such as as epicatechin, hesperidin and quercetin, and stilbenes, such as resveratrol.

We found two effective senolytic combinations with dasatinib in addition to dasatinib and quercetin by using resveratrol and ellagic acid, opening new perspectives for additional preclinical and clinical testing focusing on lung diseases associated with aging.

## 2. Results

### 2.1. Doxorubicin (DOXO)-Induced Senescence in MRC-5 and A549 Cell Lines

Exposure of lung cell lines to DOXO induces more evident morphological changes in the MRC-5 fibroblast than in A549 cells. After 20 days of DOXO exposure, MRC-5 displays apparent morphological alterations, such as enlarged, multinucleated, and vacuolated cells different from elongated control cells. A cell cycle arrest was evident due to the absence of proliferation observed (Figure 1A,B). For MRC-5, 95.6 ± 0.6% of cells were SA-βgal positive (Figure 1E), produced a high amount of IL-6 in the 24 h-supernatant (Figure 1F), and had an increased CDKN2A and CDKN1A gene expressions, which encode proteins p16^INK4A^ and p21, respectively (Figure 1G,H) after DOXO exposure. No β-gal positive cells were observed in MRC-5 at passage 22 (Figure 1E). The A549 cell line exposure to the same DOXO protocol also resulted in 94.7 ± 1.1% of SA-βgal positive cells (Figure 1I). However, morphology alterations were not evident, and cells continued with high proliferative capacity that meant that it became impossible to extend the protocol due to cellular confluency (Figure 1C,D). In A549 cells exposed to DOXO, the release of IL-6 in the 24-hour supernatant was significantly increased (Figure 1J). The expression of gene CDKN1A was increased after DOXO treatment, but no gene expression of CDKN2A was registered in A549 (Figure 1K,L).

### 2.2. Senolytic Activity of Drug and Polyphenol Combinations

Initially, cell viability assessment assays were carried out with different concentrations of drugs and isolated polyphenols in control MRC-5 cells (Appendix A). Then, the highest concentrations of the drugs that did not show a significant cytotoxic effect upon control MRC-5 cells were combined with the highest non-cytotoxic concentrations or with the maximum tested concentration of 1 mM of the different polyphenols. The addition of non-toxic concentrations of drugs with non-toxic concentrations of polyphenols was assayed for cytotoxic activity in control MRC-5 cells and those with DOXO-induced senescence. The results showed that senescent cells were resistant to death induced by drugs combined with polyphenols (Figure 2). No senolytic activity was observed for aspirin, azithromycin, FK-506 (or tacrolimus), metformin, or rapamycin in combination with nine polyphenols. However, we observed senolytic activity when dasatinib (50 µM) was combined with resveratrol (100 µM) and ellagic acid (50 µM) in addition to the expected activity of dasatinib (50 µM) and quercetin (100 µM). Dasatinib plus quercetin reduced the cellular viability by 49.2% in senescent MRC-5, while dasatinib plus ellagic acid and dasatinib plus resveratrol reduced the viability by 43.3% and 43.9%, respectively. Notably, the senolytic activity of dasatinib plus ellagic acid was achieved with half the concentration necessary for the same activity from resveratrol and quercetin at a molar equivalent (Figure 3). Isolated drug effects upon control and senescent MRC-5 cells can be observed in Figure 2 in the absence of polyphenols, and isolated polyphenol effects upon control and senescent MRC-5 cells can be verified in Appendix A. No senolytic effect was observed.

### 2.3. Evaluation of Senolytic Mechanism

As shown in Figure 3, combinations of dasatinib with polyphenols increase the percentage of apoptotic and necrotic senescent MRC-5 cells. The number of viable senescent MRC-5 cells was reduced by 40%, 63%, and 52% by the addition of dasatinib plus ellagic acid, quercetin, or resveratrol, respectively, confirming the senolytic activity of drug combinations. The percentages of apoptotic senescent MRC-5 cells were 69.7%, 79.7%, and 76.3% after 24 h of incubation with dasatinib plus ellagic acid, quercetin, and resveratrol, respectively, indicating that apoptosis, not necrosis, is the primary process of cell death. The protein expression of BCL-2 was decreased after treatment with dasatinib and polyphenols, but the combinations of dasatinib plus quercetin and dasatinib plus ellagic acid were more inhibitory than dasatinib plus resveratrol (*p* < 0.05; Figure 4). The protein levels of BCL-xl were not altered in a significant way after the treatment of MRC-5 senescent cells with dasatinib and polyphenols. The caspase activity was significantly increased after treatment of MRC-5 senescent cells with dasatinib plus quercetin, resveratrol, or ellagic acid at the same level (Figure 4).

### 2.4. Inflammatory Markers Released during Senolysis

We also measured the IL-6 and IL-1β released by MRC-5 cells during incubation with dasatinib and polyphenols at senolytic doses. As shown in Figure 5, the combinations of dasatinib plus ellagic acid and dasatinib plus quercetin increased these cytokines in the supernatant during senolysis. However, the released cytokines during senolysis were not altered by the combination of dasatinib plus resveratrol.

### 2.5. Senolytic Activity of Dasatinib and Polyphenols in A549 Cells

As shown in Figure 6, no senolytic effect was observed when the senolytic concentrations of dasatinib combined with ellagic acid, quercetin, or resveratrol used in MRC-5 assays were added to the senescent lung epithelial A549 cell line for 24 h. The cytotoxicity was reduced in senescent cells compared to control A549 cells, reinforcing the presence of resistance to apoptosis in senescent A549 cells.

## 3. Discussion

The first generation of senolytics was developed from agents to treat cancer and targeted cell survival pathways, such as proteins from the B-cell lymphoma family 2 (BCL-2) and tyrosine kinases. Thus, the combination of dasatinib, an Src tyrosine kinase inhibitor used as a second-line treatment for resistant chronic myelogenous leukemia, and quercetin, a flavonoid and nonspecific kinase inhibitor with subsequent actions on apoptotic pathways, emerged as a senolytic option [24], triggering clinical studies. A phase I study with a small number of participants with confirmed IPF received dasatinib plus quercetin (100 mg and 1250 mg, respectively) or a placebo for three consecutive days for three weeks. Although the treatment was demonstrated to be safe, evaluating its effectiveness was impossible due to limitations of the short and intermittent treatment, the small number of participants, and an inadequate distribution of patients who were using anti-fibrotic medications [31]. A pilot open-label of this study initially demonstrated improvements in physical function evaluated as 6-minute walk distance, 4 m gait speed, and chair-stand time in participants with IPF [23]. In a phase I study, participants with diabetic kidney disease received dasatinib and quercetin (100 mg and Q 1000 mg) for three days, and a reduction in senescent adipocytes (p16 ^ink4a^ and β-gal positive cells) and serum senescence markers, such as IL-1α, IL-2, IL-6, IL-9, and MMP-2, 9, and 12, was registered [32]. Finally, another phase I study with Alzheimer participants who received dasatinib and quercetin (100 mg and Q 1000 mg) for only two days provided preliminary data on the safety of the treatment and showed that the combination crosses the blood–brain barrier [33], encouraging the initiation of phase II studies. The optimism that determined the execution of these phase I studies, and others that are still ongoing, was based on promising preclinical results. In a model of bleomycin-induced lung fibrosis in mice, dasatinib plus quercetin (5 mg·kg^−1^ and 50 mg·kg^−1^) orally for three days improved lung function in terms of exercise capacity, lung compliance and reduced lung fibrosis, and senescence markers, such as as IL-6, TNF-α, MMP-3, MMP-12, Col1A1, TGF-β, and MCP-1 [9].

We described in our work the possibility of additional drug combinations and natural polyphenols for future development as senolytics. Our results demonstrated that dasatinib with resveratrol, a stilbene, and dasatinib with ellagic acid, a dilactone of hexahydroxydiphenic acid, have potential as senolytics. Resveratrol can induce growth inhibition and apoptosis in tumor cell lines at 70–150 μM [34,35] and exert beneficial effects in aging models by its pharmacological activation of SIRT1 [36]. However, a senolytic activity screening demonstrated a negative result [37]. We confirm this result, because only resveratrol showed no senolytic effect at the doses tested in our work. We only detected senolytic activity when resveratrol was combined with dasatinib.

Ellagic acid also has been demonstrated to have potent anti-tumor activity by increasing apoptosis through increasing the Bax/Bcl-2 ratio induced caspase-3 [38,39] in a range of 10–100 μM and has been identified as an anti-aging possibility by results obtained in vivo [40]. No data about senolytic activity were registered in the literature. Of note, other natural polyphenols assayed in our work had previously demonstrated anticancer and anti-aging activity, and they were not senolytics, either alone or combined with dasatinib, such as as chlorogenic acid [41,42]. This indicates a specific senolytic activity of quercetin, resveratrol, and ellagic acid when combined with dasatinib.

The three combinations increased apoptosis in MRC-5 senescent cells by decreasing BCL-2 expression and increasing caspase-3/7 activity. Ellagic acid was assayed at 50 μM, whereas quercetin and resveratrol were at 100 μM. The ability to induce caspase-3 was previously reported with ellagic acid, as cited above, and also with resveratrol, which showed the ability to induce caspase-3 activation in U937 and HCT-116 cell lines in vitro [43,44]. However, some studies report that resveratrol can inhibit caspase activation and exhibit anti-apoptotic activity [45,46].

Interestingly, during senolysis induced by dasatinib and quercetin and dasatinib and ellagic acid in senescent MRC-5 cells, higher amounts of IL-6 and IL-1β were detected in the supernatant of cell culture. During apoptosis, anti-inflammatory factors, such as IL-10, transforming growth factor (TGF)-β, and annexin I, are released to avoid stimulation of phagocyte cells [47]. Nonetheless, necrotic cells can release cytosolic factors that activate other immune cells, such as IL-6 [48]. However, the number of necrotic cells was not increased by senolysis induced by dasatinib and quercetin/ellagic acid. The most prominent cytokine of SASP is IL-6 [49]. During senolysis by dasatinib and resveratrol, we did not observe any increase in IL-6 in the supernatant of cell culture, nor of IL-1α, as detected during senolysis induced by dasatinib and quercetin/ellagic acid. It is well documented that resveratrol causes anti-inflammatory activity by inhibiting NF-κB and counteracting cytokine production by immune cells [50,51]. A possibility is an anti-inflammatory or senomorphic activity inhibiting SASP release in parallel to senolytic action only observed with resveratrol in our culture system of MRC-5 senescent cells. Fisetin is an example of a natural compound that has demonstrated senolytic and senomorphic activity [22].

The senolytic activity of dasatinib and polyphenols assayed was only observed in senescent fibroblasts and not in another cell line, namely A549 (epithelial cells derived from human lung carcinoma). Our data suggest an apparent selective effect of combinations of dasatinib with polyphenols on senescent fibroblasts. The senolytic ABT-263 (navitoclax) selectively induced the apoptosis of DOXO-induced senescent HUVECs but not EA.hy926 cells, confirming that it may have sensibility differences from senescent cells to senolytics. In the same report, dasatinib, quercetin, and fisetin did not demonstrate selectivity [52]. It is worth mentioning that the proliferative behavior after exposure to DOXO differed between MRC-5 and A549, indicating differences in the senescence process between the lineages. The CDKN2A gene expression associated with p16^INK4A^ protein expression was absent in the A549 cell line, and it was a mechanism present in the senescence induction in MRC-5 cells [53]. Whether this selective effect of combinations of dasatinib with polyphenols on senescent fibroblasts is positive or negative or happens in vivo could just be a variation in the ability to express apoptotic proteins relative to the cell lines used in this work, and clinical studies that are being conducted with the dasatinib and quercetin combination will bring us the answer. Our in vitro study does not allow us to establish a safe and effective dose that can be translated into clinical studies. However, all compounds in this work have already been subjected to clinical studies and present safe doses for human use have been determined. If we consider the phase 1 and 2 clinical trials of dasatinib plus quercetin that have used dasatinib at a dose of 100 mg orally plus quercetin at 1250 mg orally, we could suggest that the combination of 1250 mg of resveratrol or 625 mg of ellagic acid to dasatinib could have efficacy. Clinical trials using resveratrol at 1500 mg/day did not demonstrate adverse effects [54]. Clinical trials are being conducted using ellagic acid 500 mg/twice a day by oral route for 12 weeks with no adverse effects reported [55].

In conclusion, the elimination of senescent cells through the use of senolytics has been considered as a plausible therapeutic strategy to treat age-related lung diseases and disorders, and we emerged with two new possibilities of senolytics, dasatinib plus resveratrol and dasatinib plus ellagic acid, to be evaluated in preclinical models comparing them to the standard option of dasatinib plus quercetin. Dasatinib and resveratrol associate anti-SASP activity with senolytic activity, while ellagic acid can be used in lower doses than quercetin, which may prove beneficial.

## 4. Materials and Methods

### 4.1. Cell Culture

The MRC-5 (human lung fibroblast cells) culture was purchased from the Rio de Janeiro Cell Bank (BCRJ 0180; OS.C.5930.21; Lot:0001246; Passage:21). The cell line was stored in cryogenic conditions in our laboratory. Cells were cultured in DMEM supplemented with 1% NEAA, 10% fetal bovine serum, 1% penicillin, streptomycin, and amphotericin solution in a humidified incubator at 37 °C under an atmosphere of 5% CO_2_. All cell culture reagents were purchased from Gibco, Carlsbad, CA, USA. A549 was also purchased from Rio de Janeiro Cell Bank (BCRJ 0033; OS.C.6429.22; Lot:001313; Passage:88) and cultured in DMEM supplemented with 10% fetal bovine serum, 1% penicillin, streptomycin, and amphotericin solution, as described. All culture reagents were purchased from Invitrogen (Carlsbad, CA, USA).

### 4.2. Senescence Induced by Doxorubicin

MRC-5 cells were seeded onto 96-well plates at a density of 3 × 10^4^ cells/well and were incubated with DOXO 0.25 or 1.5 µM (purity ≥ 95%, Sigma-Aldrich, Saint Louis, MO, USA) for 24 h. After 24 h, the medium was changed, and no more DOXO was added. Cells were cultivated as described above, and experiments were performed after 6 and 20 days for protocol standardization (Appendix A). Control cells were MRC-5 passage 22. Senescence was also induced with doxorubicin 1.5 µM in A549 cells as described. The A549 experiments were carried out after 10 days in culture. Senescence induction was characterized by SA-βgal activity, IL-6 release, CDKN2A, and CDKN1A gene expression to determine the optimal experiment time after DOXO exposure for each cell line. For SA-βgal activity, cells were fixed with 3.7% formaldehyde in phosphate-buffered saline (PBS) for 3 min. After washing with PBS, cells were stained with a solution containing 1 mg/mL of X-gal (Invitrogen), 40 mM citric acid/sodium phosphate buffer (pH 6.0), 5 mM potassium ferricyanide (Merck, Saint Louis, MO, USA), 5 mM potassium ferrocyanide (Merck), 150 mM NaCl, and 2 mM MgCl_2_ for six hours at 37 °C. Cells were washed with PBS and photographed in five random fields for cell count. For gene expression analysis, mRNA was purified using a RNeasy Plus kit (Qiagen, Hilden, Germany). Real-time PCR was performed in a QuantStudio 1 real-time PCR system (Applied Biosystems, Waltham, MA, USA). cDNA was synthesized using the High-Capacity cDNA Reverse Transcription kit (Thermo Fisher, Waltham, MA, USA). Quantitative RT-PCR was performed using SybrGreen Master Mix (Thermo Fisher) and the primers described in the Appendix A. The expression of b-actin rRNA was used as an endogenous control for data normalization. The results were analyzed using the 2^−ΔCt^ relative quantification method.

### 4.3. Screening for Senolytic Activity

Azithromycin monohydrate (0.1–10 µM; purity ≥ 96%, NC Farma, Hortolândia, Brazil), rapamycin (2–200 nM; purity ≥ 95%, Sigma-Aldrich), FK-506 (0.1–10 µM; purity ≥ 98%, NC Farma), and dasatinib (0.5–100 µM; purity ≥ 98%, Sigma-Aldrich) were dissolved in DMSO followed by dilutions in DMEM. The DMSO final concentration in cell experiments was less than 0.1%. Metformin (0.5–50 mM; purity ≥ 98.5%, NC Farma) and aspirin (0.01–10 mM; purity ≥ 99%, Sigma-Aldrich) were dissolved in DMEM. Control MRC-5 cells were incubated for 24 h with a range of concentrations for each drug, and viability was evaluated by MTT assay. The culture medium was removed, and the MTT solution (0.5 mg. mL−1 in PBS) was added for two hours at 37 °C under an atmosphere of 5% CO_2_. The formazan crystals were solubilized in isopropanol for 10 min, and absorbance was read at 540 nm (Berthold technology; Bad Wildbad, Germany). Each experiment was performed in triplicate with one repetition. The cell viability was also evaluated in control cells incubated during 24 h with a concentration range of the polyphenols caffeic acid (1–1000 µM; purity ≥ 98%, Sigma-Aldrich), chlorogenic acid (1–1000 µM; purity ≥ 95%, Sigma-Aldrich), ellagic acid (1–1000 µM; purity ≥ 95%, Sigma-Aldrich), epicatechin (1–1000 µM; purity ≥ 98%, Sigma-Aldrich), ferulic acid (1–1000 µM; purity ≥ 99%, Sigma-Aldrich), gallic acid (1–1000 µM; purity ≥ 97.5%, Sigma-Aldrich), hesperidin (1–100 µM; purity ≥ 80%, Sigma-Aldrich), quercetin (1–1000 µM; purity ≥ 95%, Sigma-Aldrich), and resveratrol (1–1000 µM; purity ≥ 99%, Sigma-Aldrich). Chlorogenic acid and gallic acid were dissolved in DMEM, and all the others were dissolved in DMSO, as previously described. The higher non-cytotoxic dose of drugs and polyphenols were combined, and another viability assay was performed with control MRC-5 as described. The same doses of drug and polyphenol combinations were used in a viability assay with senescent MRC-5. Combinations of drugs and polyphenols that showed senolytic activity in MRC-5 cells were also tested using control and senescent A549 cells, according to the methodology described.

### 4.4. Apoptosis Detection Assays

Apoptosis was evaluated using dead cell apoptosis kits with Annexin V for a flow cytometry kit (Invitrogen). Briefly, after incubation with the combinations of dasatinib and polyphenols, cells were washed with PBS and harvested. Cell suspensions (1 × 10^5^ cells) were incubated with FITC–Annexin V and propidium iodide 1 ug/mL in buffer containing 10 mM HEPES, 140 mM NaCl, 2.5 mM CaCl2, pH 7.4 at room temperature for 15 min. Cell suspensions were diluted 5× in assay buffer and maintained at 4 °C. Stained cells were analyzed by flow cytometry, measuring fluorescence emission at 530 nm (FL1) and >575 nm (FL3). Living cells showed a low level of fluorescence, apoptotic cells showed green fluorescence, and dead cells showed red and green fluorescence.

Caspase activity was evaluated by the fluorometric method using an EnzChek^®^ Caspase-3 Assay Kit (Invitrogen). Briefly, after incubating the cells with the combinations of dasatinib and polyphenols, the cells were washed with PBS, harvested, and frozen at –80 °C until analysis. After thawing, cells were lysed by resuspension in buffer containing 200 mM TRIS, pH 7.5, 2 M NaCl, 20 mM EDTA, and 0.2% TRITON X-100, followed by a freeze–thaw cycle. The sample was diluted in a working buffer containing 50 mM PIPES, pH 7.4, 10 mM EDTA, 0.5% CHAPS, and 160 mM dithiothreitol, and centrifuged to pellet the cellular debris. The supernatant was transferred to a microplate and incubated with 50 μM Z-DEVD–R110 substrate for 30 min at room temperature. Fluorescence readings (excitation/emission ~496/520 nm, respectively) were taken continuously every 5 min for 30 min. The variation in fluorescence observed for each sample was corrected for the quantity of total proteins in the same sample. Total protein measurement was performed using the Bradford assay (Thermofisher).

### 4.5. BCL-2/BCL-xl, IL-6, and IL-1β Quantification

BCL-2 and BCL-xl proteins were quantified in cell homogenate using ELISA kits (Human total BCl-2 and Human total BCL-XL duoset ELISA—R&D Systems, Minneapolis, MN, USA). Cells were harvested using cell lysis buffer (150 mM NaCl, 50 mM Tris-HCl, 1 mM EDTA, 0.1% Triton X-100, 0.1% SDS, and 0.1% sodium deoxycholate) added with 1% protease and phosphatase inhibitors cocktail (Merck). Il-6 and IL-1β were measured in a culture medium obtained from control and senescent cell cultivation after 24 h without FBS, using EIA kits (BD OptEIA human IL-6 kit and IL-1β, San Diego, CA, USA).

### 4.6. Statistical Analysis

All data are expressed as mean ± S.E.M. Data comparisons were performed using a one-way analysis of variance followed by Student’s t-test or Bonferroni’s multiple comparison test. An associated probability (*p* value) of less than 0.05 was considered significant.

## Figures and Tables

**Figure 1 pharmaceuticals-17-00070-f001:**
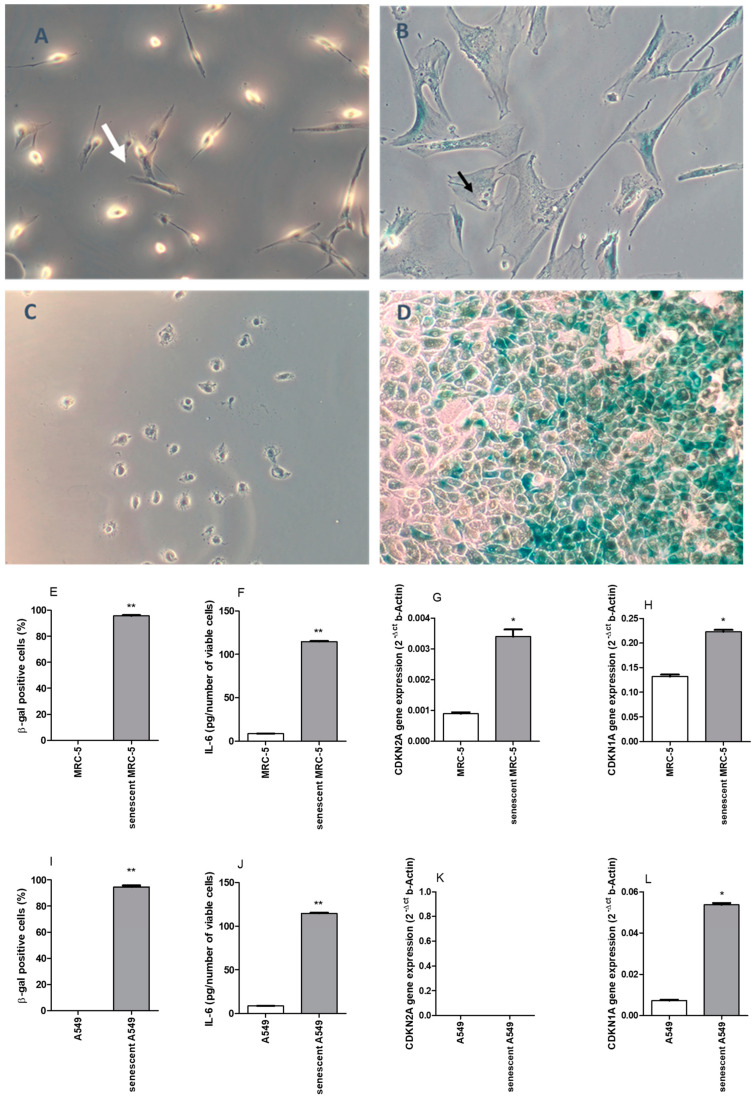
MRC-5 control (**A**) and MRC-5 20 days after 24 h of doxorubicin exposure (**B**), and A549 control (**C**) and A549 10 days after 24 h of doxorubicin exposure (**D**). The white arrow shows a normal MRC-5 fibroblast with characteristic morphology. The black arrow shows an enlarged and multinucleated senescent MRC-5 fibroblast. Blue cells after six hours of X-gal staining as described in the Methods section. 100×. Percentage of β-gal positive cells (**E**), IL-6 measurement in the supernatant (**F**), CDKN2A gene expression (**G**), and CDKN1A gene expression (**H**) for control and DOXO-treated MRC-5 cells. Percentage of β-gal positive cells (**I**), IL-6 measurement in the supernatant (**J**), CDKN2A gene expression (**K**), and CDKN1A gene expression (**L**) for control and DOXO exposure A549 cells. The experiments were carried out in triplicate. * *p* < 0.05 and ** *p* < 0.01 when comparing senescent (DOXO-treated) cells with control MRC-5 cells (non-treated).

**Figure 2 pharmaceuticals-17-00070-f002:**
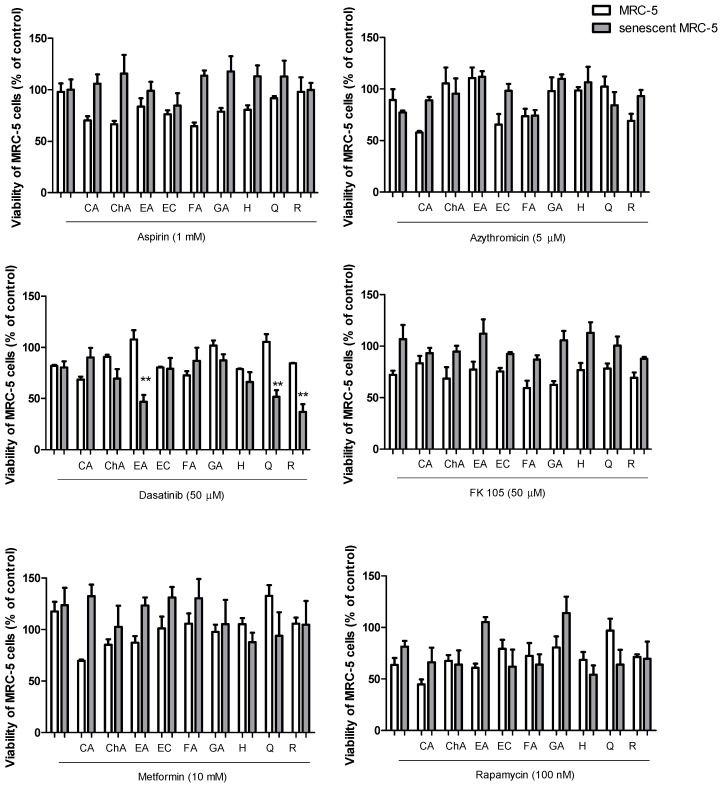
Cytotoxic activity in MRC-5 cells or senescent MRC-5 cells with aspirin (10 µM), azithromycin (5 µM), dasatinib (50 µM), FK506 (50 µM), metformin (10 mM), or rapamycin (100 nM) isolated or combined with caffeic acid (CA, 1000 µM), chlorogenic acid (ChA, 1000 µM), ellagic acid (50 µM), epicatechin (EC, 1000 µM), ferulic acid (FA, 1000 µM), gallic acid (GA, 100 µM), hesperidin (H, 100 µM), quercetin (Q, 100 µM), or resveratrol (R, 100 µM). The experiments were carried out in triplicate with one repetition. ** *p* < 0.01 when comparing senescent cells with control MRC-5 cells.

**Figure 3 pharmaceuticals-17-00070-f003:**
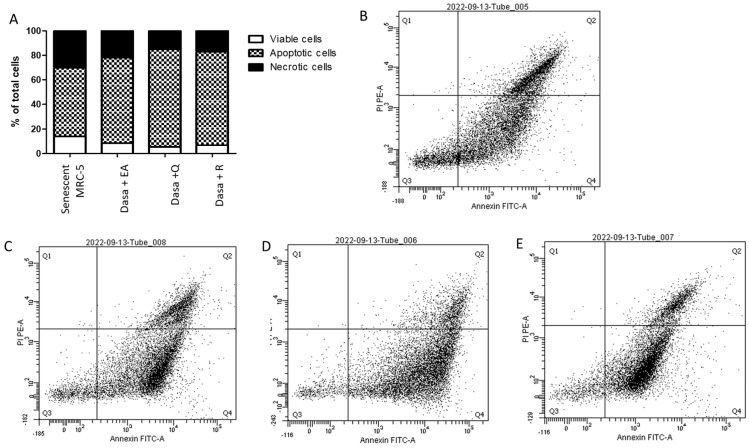
Viable (Q3 gating), apoptotic (Q4 gating), and necrotic (Q2 gating) senescent MRC-5 cells (**A**). Dot plot of senescent MRC-5 cells (**B**), senescent MRC-5 cells after 24 h of incubation with dasatinib combined with ellagic acid (Dasa + EA; 50 µM, (**C**)), quercetin (Dasa + Q; 100 µM, (**D**)), and resveratrol (Dasa + R; 100 µM, (**E**)) labeled with Annexin V conjugate to identify apoptotic cells and with propidium iodide to identify necrotic cells. The experiments were carried out in triplicate and pooled for one cytometric analysis.

**Figure 4 pharmaceuticals-17-00070-f004:**
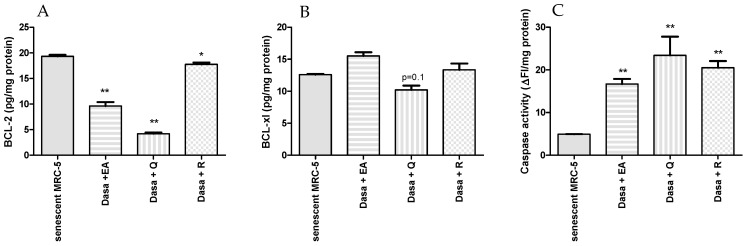
Expression of protein Bcl-2 (**A**), Bcl-xl (**B**), and caspase activity (**C**) in senescent MRC-5 cells and senescent MRC-5 cells after 24 h of incubation with dasatinib combined with quercetin (Dasa + Q; 100 µM), resveratrol (Dasa +R; 100 µM), or ellagic acid (Dasa + EA; 50 µM). The experiment was carried out in duplicate with one repetition. * *p* < 0.05 and ** *p* < 0.01 when compared to senescent MRC-5 cells.

**Figure 5 pharmaceuticals-17-00070-f005:**
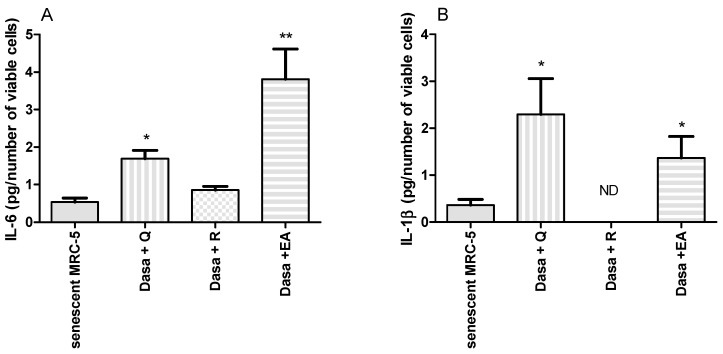
IL-6 (**A**) and IL-1β (**B**) released by senescent MRC-5 cells, and senescent MRC-5 cells after 24 h of dasatinib combined with quercetin (Dasa + Q; 100 µM), resveratrol (Dasa +R; 100 µM), or ellagic acid (Dasa + EA; 50 µM). The experiment was carried out in triplicate with one repetition. * *p* < 0.05 and ** *p* < 0.01 when compared to senescent MRC-5 cells.

**Figure 6 pharmaceuticals-17-00070-f006:**
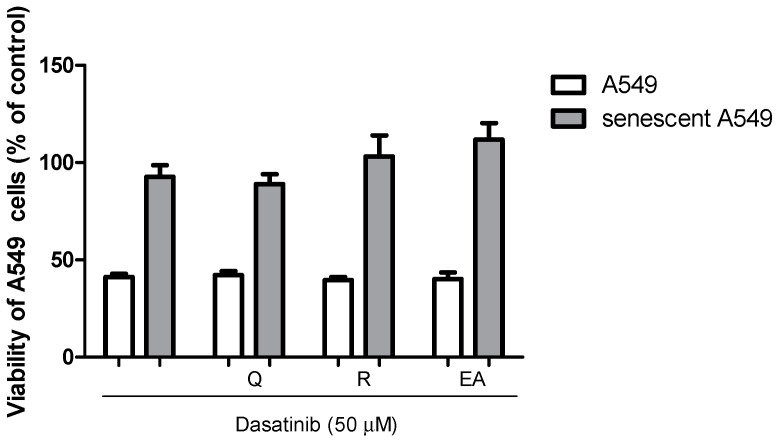
Cytotoxic activity in A549 cells or senescent A549 cells by dasatinib (50 µM) isolated or combined with ellagic acid (50 µM), quercetin (100 µM), or resveratrol (100 µM). The experiment was carried out in triplicate with one repetition.

## Data Availability

Data is contained within the article and Appendix A.

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
