# Peer review of "Researching New Drug Combinations with Senolytic Activity Using Senescent Human Lung Fibroblasts MRC-5 Cell Line"

_pharmaceuticals, 2024, doi:10.3390/ph17010070_

Round 1

Reviewer 1 Report

Comments and Suggestions for Authors

The manuscript proposed by Gambero and co-wokers is interesting since it provides novel insights on combination of senolytics that can used in potential clinical trials in the context of lung diseases that represent an important public health issue. While the topic is interesting and overall the experimental strategy well designed, the presentation of the results needs significative improvement prior to publication.

Some major comments :

Without any exception, please better detail all the containing of the figure, for exemple Figure 3 is not discussed at all. Figures 4 and 5 can be gathered to have better impact. The dose used for the testing are not introduced at all. Please better justify the concentrations used. 

Moreover, the results are very superficially discussed, please better put in perspective the results. 

Minor points 

Figure 2 : the concentration of metformin seems not correct, please check

line 139 : it is referred to Figure 4 but results presented are suitable with  Figure 3

Author Response

Some major comments :

Without any exception, please better detail all the containing of the figure, for exemple Figure 3 is not discussed at all. Figures 4 and 5 can be gathered to have better impact. The dose used for the testing are not introduced at all. Please better justify the concentrations used.

Moreover, the results are very superficially discussed, please better put in perspective the results. 

R: All dose-response curves carried out for the definition of dose for subsequent assays were included in Supplementary material. We agree that it will facilitate understanding the chosen doses' rationale. Figures 4 and 5 were gathered. We improved the description of Figure 3 and the discussion of the results.

Minor points 

Figure 2 : the concentration of metformin seems not correct, please check

R: It was corrected.

line 139 : it is referred to Figure 4 but results presented are suitable with  Figure 3

R: It was corrected.

Reviewer 2 Report

Comments and Suggestions for Authors

The manuscript entitled “Researching new drug combinations with senolytic activity using senescent human lung fibroblasts MRC-5 cell line” by Marian Carolina Ximenes de Godoy et.al describes the study of new combination of senolytic drugs against senescent MRC-5 cell line. In this study they have identified dasatinib and ellagic acid and dasatinib and resveratrol as potential drug combinations alternative to dasatinib and quercetin in the senotherapeutic field

Comments;

  1. Rationale behind choosing the drug combination was not clear. Authors has briefs mentioned as polyphenols cocktails that could have senolytic activity. How they came across azithromycin, rapamycin, metformin, FK-506, aspirin drugs (which are not related with each : for example scaffold structure or mode of action) was not described.
  2. Kindly mention the chemical structure for the compounds in the study
  3. What is the purity of these drugs? and the formulation composition for all the drugs? Please include the catalog details, purity of the drugs in the method sections.
  4. The identified drug combination dasatinib and ellagic acid and dasatinib and resveratrol how specific towards the senescent cells compared to non-senescent cells
  5. Western blot image to validate the expression level of senescent markers will be help in the given drug combinations ex: p16 and BCL-2 immunoblots in the presence and absence of drugs combinations.
  6. Figure 2. What are the left columns in the bar graphs. The description was missing
  7. Please mention the abbreviations at the first instance.

Ex; - full form of DOXO in line 84

  1.  For the bar graph please perform the statistics and highlight the significance for each  set. Current representation is not acceptable
  2. What is the effect of Quercetin, resveratrol and ellagic acid alone without dasatinib in the senescent vs nonsensescnet cells? please include the details in the supporting data
  3.  The formulations provided in the current study shows the concentation in micromolar range. Who it is safe for clinical studies or their toxicity profiles.
  4. Figure 3C, please mention the legends on the plots and describe how gatting is done.
  5.  What is the cell cycle status of the cells ? or fate of the cells after Doxorubicin treatment
  6. Has authors verified the levels of the gamma H2AX IF after 24 hr treatment With and without DOXO and the drug combinations?
  7.  Its not clear weather DOXO was still used during the senolytic drugs treatment of DOXO induced senescent cells

Comments on the Quality of English Language

Minor changes in english is required

Author Response

Comments;

  1. Rationale behind choosing the drug combination was not clear. Authors has briefs mentioned as polyphenols cocktails that could have senolytic activity. How they came across azithromycin, rapamycin, metformin, FK-506, aspirin drugs (which are not related with each : for example scaffold structure or mode of action) was not described.

R: Although they do not share a mechanism of action or chemical structure, azithromycin, rapamycin, and aspirin were previously described as senolytics, and metformin and FK-506 as senomorphics; thus, they were chosen. We included more data about it in the Introduction section.

2. Kindly mention the chemical structure for the compounds in the study.

R: In the Introduction section, we included a brief classification of polyphenols selected for this study.

3. What is the purity of these drugs? and the formulation composition for all the drugs? Please include the catalog details, purity of the drugs in the method sections.

R: Azithromycin, FK-506, and metformin were donated by NC Pharma Brazil. All other drugs and polyphenols were purchased from Sigma-Aldrich. The purity and source were included in the methods section.

4. The identified drug combination dasatinib and ellagic acid and dasatinib and resveratrol how specific towards the senescent cells compared to non-senescent cells

R: We calculated each combination's effect (%) and added to the Result section text. A table with the data used from this calculation was included in the Supplementary material.

5. Western blot image to validate the expression level of senescent markers will be help in the given drug combinations ex: p16 and BCL-2 immunoblots in the presence and absence of drugs combinations.

R: We chose to work with ELISAs as a quantitative method rather than semiquantitative WB. BCL-2 quantification data were presented in the article. The quantification of p16 would indicate the cell cycle arrest motif associated with senescence and was included as part of the senescence protocol standardization experiments we employed (gene expression analysis).

6. Figure 2. What are the left columns in the bar graphs. The description was missing.

R: The columns on the left are covered by the bar below the graph indicating the drug used in that experiment. Therefore, the columns indicate cell viability in the presence of the drugs alone (without polyphenols), with the white bar relating to control cells and the gray bar relating to senescent cells. A better description was added to the Result section.

7. Please mention the abbreviations at the first instance.

Ex; - full form of DOXO in line 84

R: I apologize for the inattention. It was included.

8. For the bar graph please perform the statistics and highlight the significance for each  set. Current representation is not acceptable.

R: The statistical analysis used in the work was elementary as it aimed to compare data from senescent cells versus control or treatment versus non-treatment situations. We seek to standardize how the level of statistical significance is identified.

9. What is the effect of Quercetin, resveratrol and ellagic acid alone without dasatinib in the senescent vs nonsensescnet cells? please include the details in the supporting data

R: No differences in the cytotoxic effect were observed using quercetin, resveratrol, ellagic acid, or other polyphenols alone. The graphic was included in the Supplementary material (Figure S4).  

10. The formulations provided in the current study shows the concentation in micromolar range. Who it is safe for clinical studies or their toxicity profiles.

R: In vitro screening does not allow for establishing safe doses that can be transferred to clinical studies. However, all compounds in this work have already been subjected to pre-clinical or clinical studies and present safe doses for human use.

We can use the clinical trials of dasatinib plus quercetin for an exercise.

A phase 2 study for Accelerated Aging in Mental Disorders is employing 8 doses of the medications - two consecutive days of dasatinib 100mg orally plus quercetin 1250mg orally on weeks 1 (ie., 2 days on, 5 days off), 2, 3, and 4. In our in vitro study, the effective dose was 50 uM for dasatinib and 100 uM for quercetin. So, we could consider that the same dose of dasatinib (100 mg orally) and 1250 mg of resveratrol could be an idea for a clinical study. Studies on the toxicity of resveratrol in humans demonstrated that this compound is well tolerated, and no adverse effect has been found with a higher dosage (5g/day). If we considered the combination of dasatinib plus ellagic acid, half dosage could be rational, i.e. 625 mg. Clinical data for ellagic acid are scarcer, but from pre-clinical data, the estimated effective human dose would therefore be 0.4–12.2 mg EA/kg BW; thus, for an average 70-kg individual, the dose is ∼30–850 mg EA/d. We added some discussion about this topic in the manuscript.

11. Figure 3C, please mention the legends on the plots and describe how gatting is done.

R: It was included in the Figure 3 legend.

12. What is the cell cycle status of the cells ? or fate of the cells after Doxorubicin treatment

R: We did not evaluate the cell cycle, but the literature indicates that senescent cells irreversibly enter the G0 state and express b-gal as a characteristic that differentiates them from quiescent cells (doi: 10.3390/ijms22063102 for reference).

13. Has authors verified the levels of the gamma H2AX IF after 24 hr treatment With and without DOXO and the drug combinations?

R: No, we did not.

14. Its not clear weather DOXO was still used during the senolytic drugs treatment of DOXO induced senescent cells

R: Doxorubicin was only present during the first 24 hs for senescence induction. All MRC-5 experiments were performed after 20 days of senescence induction without doxorubicin. We improved the method description.

Comments on the Quality of English Language

Minor changes in english is required – The manuscript was revised.

Reviewer 3 Report

Comments and Suggestions for Authors

In this manuscript the authors examined the effects of multiple drug combinations on doxorubicin-induced senescence of human lung fibroblasts with the aim of finding candidates for senolytic treatment. Although the topic of the study is interesting, the manuscript contains multiple errors and requires multiple changes before being considered for publication.

Major comments:

1)    Chapter 2.1

-     It is hard to follow the text at the beginning of chapter 2.1 since Figure 1B is mentioned before Figure 1A, and Figure 1D is not assigned in the text.

-     The quantifications of B-gal and IL-6 expression and their statistics described in the text should be presented as graphs in Figure 1.

-     The authors need to address the rationale for choosing 1.5 μM doxorubicin for senescence induction in studied cells. Given that different cells can have different dynamics of doxorubicin response, the dose-dependent curve should be performed and added in supplementary data.

-     The protocol applied for treatments needs to be better described. It is not clear if the 1.5 μM treatment of doxorubicin was performed one time, and after 24h the media was exchanged for media without doxorubicin, or if the cells were kept under constant doxorubicin regimen for a longer time.

2)    Chapter 2.2

-     How do the authors choose the doses of studied drugs? It appears from Material and Methods section that they evaluated a range of concentrations for each drug. In this case, the obtained dose response curve for each drug should be shown in the supplementary data.

-     The results presented in the Figure 3 need to be elaborated in the text, specifically the authors should discuss the percentage of apoptotic cells observed in senescent cells and their changes after drug treatments.

3)    Chapter 2.3

-     In the first sentence of Chapter 2.3, the authors state: “As shown in Figure 4, combinations of dasatinib with polyphenols increase the percentage of apoptotic and necrotic senescent MRC-5 cells.” However, Figure 4 does not illustrate the results of the apoptotic assay. Please clarify.

4)    Chapter 2.4

-     Chapter 2.4 is as follows: “We also measured the IL-6 and IL-1β released by MRC-5 cells during incubation with dasatinib and polyphenols at senolytic doses. As shown in Figure 6, the combination of dasatinib plus resveratrol reduced these cytokines. However, the amount of cytokines produced during senolysis was not altered by the combinations of dasatinib plus quercetin or dasatinib plus ellagic acid.”  

Conversely, the results presented in Figure 6 illustrate no changes in production of IL-6 and IL-1β for the combination of dasatinib plus resveratrol, and show significant increase for combinations of dasatinib plus quercetin and dasatinib plus ellagic acid. The authors must solve this issue to interpret the data correctly.

Minor points:

1)    The legend for Figure 1 is missing “D” in description.

2)    In Figure 2 is missing descriptions of abbreviations for multiple drugs

3)    “Dasatinib” is labeled either as “D” or as “Dasa”. It should have the same abbreviation across the manuscript.

4)     “Young cells” should be exchanged on “Untreated cells”

Comments on the Quality of English Language

The manuscript will benefit from grammar check.

Author Response

Major comments:

1)    Chapter 2.1

-     It is hard to follow the text at the beginning of chapter 2.1 since Figure 1B is mentioned before Figure 1A, and Figure 1D is not assigned in the text.

R: It was modified.

-     The quantifications of B-gal and IL-6 expression and their statistics described in the text should be presented as graphs in Figure 1.

R: We transformed the data cited in the text into graphs and added the p16 and p21 gene expression evaluated during the standardization process.

-     The authors need to address the rationale for choosing 1.5 μM doxorubicin for senescence induction in studied cells. Given that different cells can have different dynamics of doxorubicin response, the dose-dependent curve should be performed and added in supplementary data.

R: It was added. We tested two concentrations and two times after incubation during the standardization of senescence induction experiments.

-     The protocol applied for treatments needs to be better described. It is not clear if the 1.5 μM treatment of doxorubicin was performed one time, and after 24h the media was exchanged for media without doxorubicin, or if the cells were kept under constant doxorubicin regimen for a longer time.

R: Doxorubicin was only present during the first 24 hs for senescence induction. All MRC-5 cell experiments were performed after 20 days of senescence induction without doxorubicin. Experiments with A549 cells were performed after 10 days, without doxorubicin. We improved the method description.

2)    Chapter 2.2

-     How do the authors choose the doses of studied drugs? It appears from Material and Methods section that they evaluated a range of concentrations for each drug. In this case, the obtained dose response curve for each drug should be shown in the supplementary data.

R: All dose-response curves carried out for the definition of dose for subsequent assays were included in a Supplementary material file. We agree that it will facilitate understanding the chosen doses' rationale.

-     The results presented in the Figure 3 need to be elaborated in the text, specifically the authors should discuss the percentage of apoptotic cells observed in senescent cells and their changes after drug treatments.

R: It was added to the Results description.

3)    Chapter 2.3

-     In the first sentence of Chapter 2.3, the authors state: "As shown in Figure 4, combinations of dasatinib with polyphenols increase the percentage of apoptotic and necrotic senescent MRC-5 cells." However, Figure 4 does not illustrate the results of the apoptotic assay. Please clarify.

R: We apologize for the inattention. The correct one is Figure 3.

4)    Chapter 2.4

-     Chapter 2.4 is as follows: "We also measured the IL-6 and IL-1β released by MRC-5 cells during incubation with dasatinib and polyphenols at senolytic doses. As shown in Figure 6, the combination of dasatinib plus resveratrol reduced these cytokines. However, the amount of cytokines produced during senolysis was not altered by the combinations of dasatinib plus quercetin or dasatinib plus ellagic acid."  

Conversely, the results presented in Figure 6 illustrate no changes in production of IL-6 and IL-1β for the combination of dasatinib plus resveratrol, and show significant increase for combinations of dasatinib plus quercetin and dasatinib plus ellagic acid. The authors must solve this issue to interpret the data correctly.

R: We agree with your criticism. We altered the Result description.

Minor points:

1)    The legend for Figure 1 is missing "D" in description. R: It was included.

2)    In Figure 2 is missing descriptions of abbreviations for multiple drugs. R: It was included.

3)    "Dasatinib" is labeled either as "D" or as "Dasa". It should have the same abbreviation across the manuscript. R: We reviewed the entire manuscript for consistency.

4)     "Young cells" should be exchanged on "Untreated cells" – R: It was altered in line 301.

Comments on the Quality of English Language

The manuscript will benefit from grammar check. – The manuscript was revised.

Round 2

Reviewer 1 Report

Comments and Suggestions for Authors

The revised version of the manuscript is significatively improved. Authors have answered to the majority of the addressed comments. Please checked typo errors. 

Author Response

Thank you for the careful review. We identified typo errors and made changes.

Reviewer 3 Report

Comments and Suggestions for Authors

The authors have sufficiently addressed the prior concerns and improved the manuscript. Thus, the paper is suitable for publication.

Author Response

Thank you for the careful review.